# Experimental quantification of hydrogen content in the Earth's core

Dongyang Huang [1,2] ✉, Motohiko Murakami [2], Stephan Gerstl[3] & Christian Liebske [2]

Earth's core has long been speculated to be the largest reservoir of hydrogen (H) on the planet. However, current estimates of its H content involve substantial uncertainties, due to the challenge of quantifying H under extreme conditions. Here, we perform superliquidus metal-silicate partitioning experiments on H using laser-heated diamond anvil cells, and combine it with atom probe tomography. The direct observation of H at silicon- and oxygen-rich nanostructures in the iron alloy indicates coupled sequestration of silicon, oxygen and hydrogen into Earth's core during its formation. With the observed molar Si/H ratio close to unity, Earth's core is estimated to contain 0.07-0.36 wt.% H, equivalent to 9-45 oceans of water. Such an amount would require the Earth to obtain the majority of its water from the main stages of terrestrial accretion, instead of through comets during late addition.

Hydrogen (H) is the most abundant element in the Solar System[1], and yet the Earth is considered cosmochemically 'dry', relative to CI carbonaceous chondrites, in regard to its hydrogen concentration[2]. Although 71% of the Earth's surface is covered by ocean, mainly made of H, it has been argued that the majority of Earth's H had been stored in the core since its formation, ~4.5 billion years ago[3–6]. Based on high-pressure metal-silicate partitioning experiments/calculations, which simulate the core-mantle distribution of H during Earth's core formation in deep magma oceans, the estimated amount of hydrogen in Earth's core involves substantial uncertainties, spanning four orders of magnitude from 10 to 10,000 parts per million by weight[4,7–12]. Thus far, high-pressure experimental data in measuring H have been challenging—relying on, and hence suffering from, the fact that their H contents were inferred indirectly from the lattice expansion induced by H[12,13], with the exception at lower pressures up to 20 GPa[9,10], which, without extrapolation, is however only applicable to Mars-sized planets (~10% $M_E$). As Earth's core-mantle differentiation likely took place in terrestrial magma oceans at pressures up to 70 GPa[14,15], direct quantification of H at pertinent conditions is essential to accurately determine the fate of hydrogen during the earliest stages of Earth's evolution.

Meanwhile, as a natural consequence of these extreme conditions, made available by recent laser-heated diamond anvil cell experiments[16–19], large amounts of O (1–17 wt.%) are found to partition into metal liquids[15,20,21]. Together with the simultaneously dissolved Si, this produces a common nanoscale Si-O-rich exsolution formed during quenching (Fig. S1), which has been observed repeatedly by a substantial body of work over the decade[15,20–27]. Nevertheless, as it introduces heterogeneity for electron probe micro-analysis, the nanoscale (tens of nanometres) Si-O-rich quench texture was commonly treated as an undesired by-product that needed to be 'overcome'[15,27].

Here, through three-dimensional compositional mapping with nanoscale spatial resolution using the atom probe tomography (APT, see 'Methods'), we are able to sample the Si-O-rich nanostructure and to make direct observation of H within it (Figs. 1–3). We find that the Si-O-rich nanostructure is, in fact, strongly coupled with the high-pressure sequestration of H in Earth's core.

## Results and discussion
### Direct observation of hydrogen within an Si-O-rich nanostructure in iron
Firstly, to reproduce the extreme pressure and temperature conditions under which the Earth's core had formed, with the presence of H, we laser-heated iron metal (the metallic core) encapsulated in hydrous silicate glass to a fully molten state (the magma ocean) at superliquidus temperatures up to ~5100 K, and at pressures up to 111 GPa in a

[1]SKLab-DeepMinE, MOEKLab-OBCE, School of Earth and Space Sciences, Peking University, Beijing, China. [2]Institute of Geochemistry and Petrology, ETH Zürich, Zürich, Switzerland. [3]Scientific Center for Optical and Electron Microscopy (ScopeM), ETH Zürich, Zürich, Switzerland. ✉e-mail: dhuang@pku.edu.cn

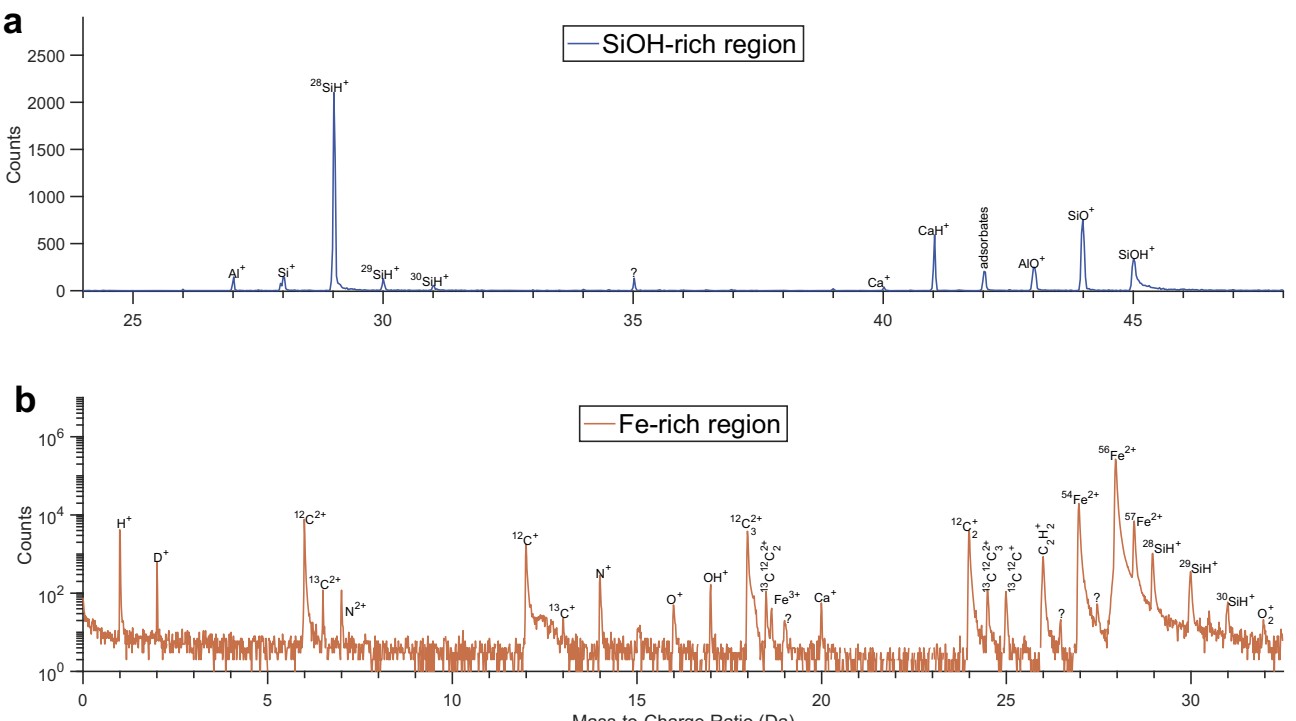

**Fig. 1 | Mass spectrum of nanostructures within the metal recovered from the laser-heated diamond anvil cell. a** Mass spectrum of the Si-O-H-rich nanostructure formed in the metal during quenching. Apart from Si, O and H, elements that are conventionally lithophile (i.e. Al and Ca) under low to ambient pressure conditions also partition into the metal, albeit at extremely low concentrations (~0.1–0.2 at.%, Supplementary Data 2 and 3). Note that the starting metal composition was pure Fe prior to the metal-silicate equilibration (see the main text). **b** Mass spectrum of the Fe-rich region in the metal. The remarkable peaks of $H^+$ and $D^+$ (or $H_2^+$) at Da = 1 and 2 are discussed in the main text. The corresponding reconstructed 3D atom map is shown in Fig. 2.

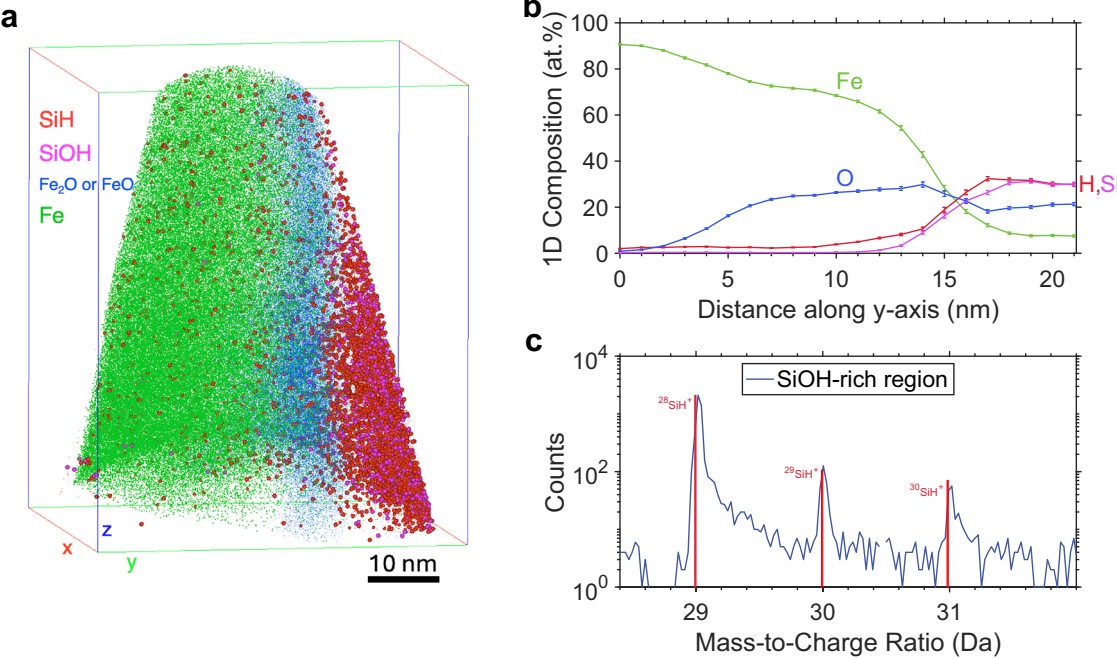

**Fig. 2 | APT analysis of the recovered metal sampling the Si-O-H-rich nanostructure. a** Full reconstructed atom map showing the Fe-rich solvent (in green), the O-rich interface (in blue), and the Si-O-H-rich exsolution (field-evaporated mainly as $SiH^+$ and $SiOH^+$ ions in red and pink) formed during rapid cooling. **b** 1D composition profile along the *y*-axis, indicating the nanoscale dynamics of the Si-O-H-rich quench texture frozen in time (see the main text). **c** Evidence for endogenous H from the Si-O-H-rich nanostructure, showing the most abundant field-evaporated $SiH^+$ (blue curve) fitting the natural abundance of Si isotopes (red vertical lines).

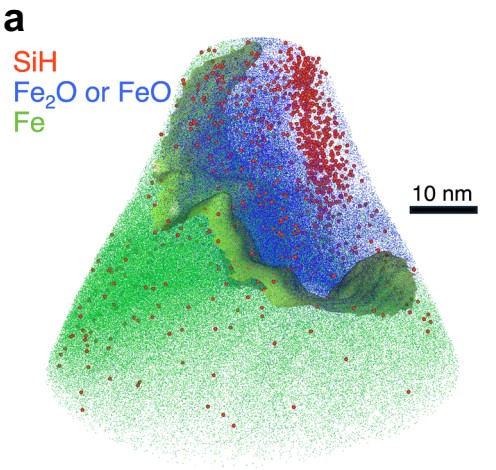

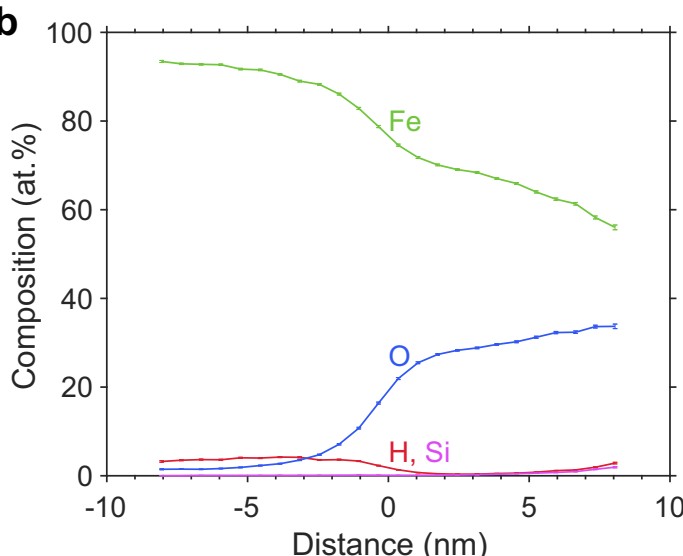

**Fig. 3 | APT analysis of another recovered metal sampling the surface of the Si-O-H-rich nanostructure. a** 3D atom map showing the Fe-rich solvent (in green), the O-rich interface (in blue), the surface of the Si-O-H-rich cluster, and the isosurface (green surface) containing 20 at.% O. **b** 1D proxigram with respect to the 20 at.% O isosurface, namely, the concentration of elements that occur at a certain distance from the defined isosurface. The H detected, up to 4 at.% in the Fe-rich matrix, may be entirely artificial due to the residual H in the APT chamber (see the main text).

diamond anvil cell (see 'Methods'). To ensure the attainment of equilibrium, the heating durations were between 6 and 10 s (ref. [15], see 'Methods'), during which time O and Si, together with H, partitioned from the silicate melt into the molten metal, which instantly exsolved as Si-O-H-rich nano-clusters upon quenching (Figs. S1 and 2, 3). Secondly, the equilibrated metal and silicate were recovered and milled, using a focused ion beam (FIB), into a sharp needle-shaped geometry, with an end diameter of ~20 nm (Fig. S2), to achieve the specimen geometry required for atom probe analysis. Thirdly, sample apexes of different regions of interest within the metal and silicate phases were placed in an ultra-high vacuum environment, arrested at cryogenic temperature (50 K), and field-evaporated ion by ion, which are quantified via time-of-flight mass spectrometry, before arriving at a 2D position-sensitive detector, enabling their back-projection reconstruction, revealing the various nanostructures.

Due to the high-field-induced stress conditions during field ionisation, fractures developed in recovered samples are common, leading to a very high failure rate during APT data collection (Supplementary Data 1). Among the few samples that survived, we managed to obtain two sets of 3D compositional data mapping the Si-O-rich exsolution and its solvent (the iron metal), which were reconstructed to provide H mapping at the atomic scale.

### The quantification of H using atom probe tomography

The analysis of H and its isotopes is known to be challenging, even with the atom probe, which in principle is capable of providing quantitative analyses across the whole periodic table[28-30]. Because the absorbed residual H stemming from the vacuum chamber may be up to 4 at.%[31], an amount so high that it accounts for nearly all the H detected, obscuring any meaningful interpretation of H data by APT. This is true in our own case, where the observed H contents in the silicates and matrix metals are up to 1 at.% (Figs. S3 and S4) and 4 at.% (Fig. 3), respectively, excluding the possibility of discerning whether or not the measured H is originated from our sample (Supplementary Data 2–5).

The unambiguous evidence for endogenous H comes from the field-evaporated SiH⁺ ions within the Si-O-H-rich nanostructure, as the isotopic ratios of Si fit robustly with their natural abundances (Fig. 2c).

This is further corroborated by statistics, because SiH⁺ is the most abundant species field-evaporated in that microstructure (Fig. 1a) and therefore contributes enormously to the final H content (>30 at.%, Fig. 2b), which cannot be explained by the residual H. On the other hand, the observation of endogenous H is supported by the relatively high abundances of protonated major rock-forming elements, e.g. CaH⁺ and SiOH⁺ at Da = 41 and 45, respectively (Fig. 1a). In the Fe-rich matrix, apart from the prominent peaks of iron isotopes (Fig. 1b), carbon, likely diffused from the diamond anvils at high temperatures[27], is also detected and resides almost entirely in the iron matrix (8.8 at.%, Supplementary Data 2). The O-rich interface (blue region in Fig. 2), between the Fe matrix and the Si-O-H-rich nanostructure, is attributed to be of quench (disequilibrium) origin, which may be absorbed by the Si-O-rich cluster to eventually form $SiO_2$ under equilibrium conditions, as the latter readily crystallises from the Fe-Si-O ternary system owing to its large liquidus field[32].

Although meticulous measurements were performed during apex preparation to avoid the fractures (Supplementary Data 1), and to sample the most interesting region-of-interest (Fig. S2), we succeeded with only one sample apex, in obtaining sufficient statistics to determine the Si-O-H-rich nanostructure (Fig. 2). A second recovered sample apex (bulk composition reported in Supplementary Data 3), mapping across the Fe-rich matrix, the O-rich interface, and a small fraction of the Si-O-H-rich cluster, is, nonetheless, consistent with the above observed compositional profiles (Fig. 3). On the other hand, based on reported diffusivities, one may expect the coupled exsolution of H, O and Si to be valid within the high-pressure and high-temperature domains relevant to Earth-sized core formation. Under probable core-forming conditions (~40-70 GPa, ~3000-4500 K)[20,27], the diffusion coefficients in liquid Fe are on the orders of $10^{-8}$ m²/s for Si and O, and $10^{-7}$ m²/s for H[33-35]. As discussed in the section 'Introduction', the Si-O-rich quench texture is an established feature at these P-T conditions in diamond anvil cell experiments; here, what we observed with APT is that H is bound to lock with Si and O. If the diffusion rates of Si and O allow for fast formation of the Si-O-rich nanostructure upon quenching, forming the Si-O-H-rich cluster under hydrous conditions would be inevitable, as the diffusivity of H is higher than those of Si and O by one order of magnitude.

## H content in Earth's proto-core

Hydrogen has long been considered a major light-element candidate[3,36–38] to account for the observed density deficit in Earth's core[39]. For decades, however, our knowledge of the exact content of H in planetary cores has been hindered by the inability to unambiguously quantify H in high-pressure samples. There are two main sources of uncertainty in estimating H content in planetary cores. First, except occasionally, H analysis is made available[9,10], the current practice of estimating H content in Earth's core involves inferring the amount of H alloyed with iron, from the lattice expansion induced by the addition of interstitial H[12,13]. This approach, however, relies on accurate determinations of lattice parameters of both iron and iron hydrides, and an implicit assumption that simultaneous dissolutions of Si and O in the iron would not induce any lattice expansion of the latter. Although the volume expansion of solid Fe induced by H is found to be approximately eight times greater than that by Si[40], the effect of the latter is not negligible. For instance, 9 wt.% Si in hcp Fe at 100 GPa may expand the lattice by about 3%[41]. Second, forward models[4,9,12] using metal-silicate partition coefficient of H ($D_H \equiv c_H^{metal}/c_H^{silicate}$, where $c$ denotes concentration of H in relevant phase) rely on the estimated water content of the silicate Earth, which further introduces uncertainties as the latter differs between studies by one order of magnitude[2,42].

Here, we propose an alternative way, independent of the conventional $D_H$ and their associated assumptions/uncertainties, to determine the H inventory in Earth's core as follows. It has been established that O partitioning into the metal is greatly enhanced at increasing pressures and temperatures (see the section 'Introduction'). An O-rich core would readily satisfy the exsolution of the Si-O-rich nanostructure, and is therefore more relevant to Earth-sized planets[43]. While the exact core composition remains highly debated, a recent review, making use of mineral physics and cosmochemical constraints, favours a slightly more O-rich core[38]. Because, under high-pressure regimes relevant to Earth-sized core formation, both H and Si almost entirely bond with O to form the Si-O-H-rich cluster during quenching, with a ~1:1 H:Si ratio (Fig. 2b and Supplementary Data 2), one may, with reasonable confidence, predict the concentration of H based on that of Si in Earth's core. While there exist large uncertainties pertaining to the estimated H content of the core (cf. Introduction), as H partitioning experiments and their quantification have been, and will continue to be, extremely challenging, the Si content of the core has been relatively well constrained to within one order of magnitude. Forward core formation modelling, compiling data from an extensive body of work on Si partitioning[14,15,17,20], inverse models that try to reconcile elasticity and geophysical observations of Earth's core[38,44], and compositional models based on meteoritic records[45] would all agree upon a generous range of 2–10 wt.% Si in the core. As such, the Earth's proto-core, i.e. the core after its formation but prior to potential mass loss to the mantle (cf. section 'Implications for the origin of Earth's water and early dynamo'), is estimated to contain 0.07–0.36 wt.% H, corresponding to the H content of 9–45 oceans of water.

Note that H becomes more siderophile with increasing pressure and temperature[4,7,9,11,12], therefore, our estimate based on experiments performed at up to 86 GPa and 5115 K (Supplementary Data 1) provides an upper limit for the water content of Earth's core. We would also like to stress that the use of the H/Si ratio to infer H content does not necessarily imply that H follows Si into the metal during metal-silicate partitioning. There are two processes involved here: metal-silicate equilibration at high temperatures, and exsolution of the Si-O-H-rich nanostructure in liquid Fe upon quenching. While the APT-observed results indicate the latter, it does not discern the specific mechanisms that occurred in the former. Compared to existing models for Earth's core composition[4,9,11,12,45], this is a somewhat less H-rich core, and requires its density deficit[39] be accounted for by a mixture of light elements, rather than a single light species, akin to that of Mars' core[46].

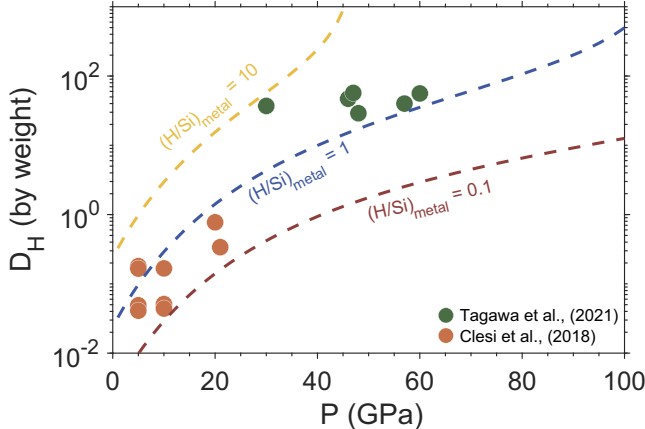

**Fig. 4 | Metal-silicate partition coefficient of hydrogen inferred from varying H/ Si ratios in the metal.** The APT-observed H/Si ratio is close to unity (see the main text), which is allowed in here to vary from 0.1 to 10, in order to quantify its effect on $D_H$. $D_H$ (dashed lines) is calculated using (i) an Si content of the core at given pressures as determined by Si partitioning model in ref. 17, and (ii) a moderate H content (120 ppm) in the primitive mantle[50].

Despite that our approach may avoid certain assumptions/ uncertainties in the other studies, it certainly involves uncertainties of its own. (i) The uncertainties in H quantification using the atom probe. As discussed in the section 'The quantification of H using atom probe tomography', the residual H from the chamber may contribute up to 4 at.% of the total measured H. Taking the face value, our measured H content in the Si-O-rich cluster (36 at.%, Supplementary Data 2) would be overestimated by 11%, which would adjust the H/Si ratio downward to 0.97. In Fig. 4, we explore the quantitative effect of varying the H/Si ratio on the H content of the core. Another uncertainty comes from the potential underestimation of H content in the Fe matrix. Although H is found to have a strong affinity with the Si-O-rich nanocluster, the exact partition coefficient of H between the Si-O-rich nanocluster and Fe matrix is unknown under these pressures, which deserves serious investigation in future studies. As such, residual H in the matrix may form iron hydride (FeHx) with Fe upon quenching, in which H is capable of escaping from solid iron during decompression. A recent study estimated the H content of Earth's core, based on lattice expansion of Fe introduced by the addition of H[12]. If one takes their surface values, i.e. 0.18–0.49 wt.% H in the core, our values of 0.07–0.36 wt.% H would be underestimated by a factor of ~2 at most, provided most of the H does not enter the Si-O-rich nanocluster. (ii) The uncertainties in the range of Si content of the core. Although it is generally considered more constrained than the H content, the estimated bulk Si concentration of the core is based on the same forward modelling assumptions. We want to emphasise that the 2–10 wt.% Si is a broad range, and serves only as a reference point. (iii) The underlying assumption is that a sufficient amount of H is present during the course of Earth's accretion. If Earth had accreted relatively 'dry', for example, from mainly enstatite-like material, there would not be enough H available to constantly attain the 1:1 H:Si ratio in the metal core. Hence, the estimated amount of 0.07–0.36 wt.% H, or 9–45 oceans of water in Earth's core, should be interpreted cautiously in light of the above sui generis uncertainties. On the other hand, had Earth accreted relatively 'wet', sufficiently so not only to attain the 1:1 H:Si ratio in the proto-core, but also to allow for the formation of FeHx following the exsolution of the Si-O-H-rich phase during cooling, H content of the proto-core may be adjusted upward to potentially overlap with the two recent estimates of 0.3–0.6 wt.%[12] and 0.18–0.49 wt.%[47].

### Implications for the origin of Earth's water and early dynamo

Consider a reasonably moderate water content in the bulk silicate Earth (including the surface ocean), which totals 2–4 oceans of water[2,42,48–50], it would make the core the largest reservoir of water on the Earth, and the bulk Earth to contain ~0.2–1 wt.% water (Supplementary Data 7). As discussed earlier, such an amount of water relies on the assumption that Earth most probably accreted water during the main stages of its accretion[5], similar to other major volatiles such as C[51] and N[27], instead of late delivery through hydrated chondritic materials. This is in line with the non-chondritic (cf. CI chondrite) H, C and N ratios in the bulk silicate Earth[2]. That the Earth accreted most of its water in situ along its formation is plausible from a dynamical point of view, with water delivered by planetesimals and planetary embryos[52], and is compatible with a hydrogen ingassing model involving interaction between a primordial atmosphere and a magma ocean[53]. It is also consistent with the idea that the Earth may have been built mainly from enstatite chondrite-like materials, which, apart from their isotopic similarities to the Earth[54,55], contain sufficient amounts of hydrogen to deliver more than three oceans of water and Earth-like H isotopic signatures[56].

The coupled sequestration of Si, O and H in core-forming metals at high pressures has important implications for geodynamics, geochemistry and the water cycle in the deep Earth. While the details of the deep-Earth water cycle could be refined with an accurate deep-time geotherm and Fe-Si-O-H phase diagram, the overarching mechanism of core-mantle water transport, driven by coupled dissolution and exsolution of Si, O and H in liquid Fe-rich alloys, remains robust. First, the strong affinity of H with the Si-O-rich nanostructure suggests that similar crystalline H-Si-O solids would form during cooling in the Fe-Si-O-H quaternary. Second, the addition of H would presumably delay the crystallisation of $SiO_2$[32] during secular cooling of Earth's core, via lowering its melting temperature. Early dynamo, if partly driven by core exsolutions, might then require an alternative power source[18,57], and may have to take account of the effect of H. Subsequently, if, depending on the cooling rate of the liquid core, H-Si-O phases crystallise and ascend to the base of the mantle, the buoyancy and latent heat produced by this crystallisation would promote convection in both the core and overlying mantle. Should it interact with deep-rooted mantle plumes[58], it may imprint the latter with primordial isotopic signatures preserved in the core since its formation[59–61]. Finally, the exsolution of H-Si-O solids would release the early core-sequestered water into the mantle, thereby reshaping mantle rheology, melting behaviour, and our understanding of Earth's deep water cycle.

## Methods

### Starting materials

The starting metal was a 10-micron-thick Fe foil (99.99+%, Goodfellow Cambridge Limited), which was cut into shards suitable for diamond anvil cell loading using a stainless steel razor blade. The starting silicate was a hydrous MORB glass synthesised at 1 GPa and 1450 °C using a 14 mm end-loaded piston-cylinder apparatus. The FeO-free MORB glass was initially doped with ~1 wt% $D_2O$, and subsequently analysed with (i) Electron probe micro-analysis (EPMA) to verify its chemical homogeneity (Supplementary Data 6), and (ii) Fourier Transform Infrared Spectroscopy to confirm its water concentration (~14,000 ppm in total, Fig. S5). Hydrogen intake from the environment (i.e. the pressure assembly) was anticipated and quantified to be 6467 ± 56 ppm, i.e. 46% of the total water content (Fig. S5).

### Laser-heated diamond anvil cell experiments

Symmetric diamond anvil cells with flat culet diameters of 250 μm or 300 μm were used to generate high pressure. Re gaskets were pre-indented to ~30 μm, and the sample chambers of ~125 μm or 150 μm in diameter were obtained by laser-drilling the gaskets. The silicate glasses were double-polished to 10 μm and laser-machined into disks smaller than the sample chamber. The iron metal was sandwiched between two glass disks, which were centred on the diamond anvil in the sample chamber. Sample assembly was compressed to a pressure of interest and then heated from both sides using two ytterbium lasers ($\lambda = 1.07$ μm, CW = 200 W, IPG Photonics). The focused laser spot size was ~20 μm in diameter. The sample was kept at the target (highest) temperature for 6–10 s, which is sufficient for the attainment of chemical equilibrium due to fast diffusion under similar conditions[15], particularly for H, as its diffusion rate in silicate melt is higher than the other elements by one order of magnitude[62]. Temperature was measured continuously by fitting the visible-to-near IR portion of the black-body radiation (500–900 nm) of the hot spot, with an estimated uncertainty of around ±5%[27]. Pressure was determined from the Raman shift of the diamond anvil before and after heating[63] and corrected for thermal pressure following $\Delta P = 2.7$ MPa/K[22]. Samples were quenched from high temperatures by switching off the laser, decompressed from high pressures, and recovered using a FIB instrument (Helios NanoLab 600i, Thermo Fisher Scientific), following previous experimental protocol[15,27].

### Atom probe tomography (APT)

APT provides three-dimensional atomic-scale mapping and compositional analysis of materials through combining point-projection-based field ion microscopy and time-of-flight mass spectrometry[64]. Samples recovered from high pressures were sharpened into needle-like tips with apex diameters of <100 nm, using the above-mentioned FIB via standard procedures[65], applying 30 kV potential with a range of milling currents (40 pA to 2.5 nA) and 5 kV final polishing to remove 30 kV implanted Ga. This geometry (Fig. S2) was made to ensure high electric fields at the tip apex, crucial for controlled field evaporation. Cameca Local Electrode Atom Probe 4000X-HR, enabled with a cryo-transfer capability[66], was used to collect the data. The specimen temperature was set to 50 K, and laser pulse energies of 120–140 pJ were applied with a pulse frequency of 125 kHz and sample final voltages of 4–7 kV. The mass-to-charge ratios of the field-evaporated ions were determined by the Time-of-Flight Mass Spectrometer. Conservative detection rates (≤0.5%) were applied to minimise high-field-induced fractures. 3D reconstructions and data analysis were completed with IVAS 3.8.

## Data availability

All data are available in the manuscript or the Supplementary Materials.

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

## Acknowledgements

D.H. is grateful to Georg Spiekermann and Mona Lüder for their help and insights regarding laser heating techniques and FTIR measurements, respectively. D.H. thanks supports from the PKU starting grant (No. 7100604887) and the National Natural Science Foundation of China (NSFC) under grant 8206100810. M.M. acknowledges support from the ETH Zürich Start-up fund PSP1-001828-000.231.

## Author contributions

Conceptualisation: D. Huang. Methodology: D. Huang, S. Gerstl. Formal analysis: D. Huang, S. Gerstl. Investigation: D. Huang, M. Murakami, S. Gerstl, C. Liebske. Supervision: D. Huang. Validation: D. Huang, S. Gerstl. Visualisation: D. Huang. Funding acquisition: D. Huang, M. Murakami. Writing—original draft: D. Huang. Writing—review and editing: D. Huang, M. Murakami, S. Gerstl, Christian Liebske.

## Competing interests

The authors declare no competing interests.
