## [Transparent Peer Review file · Nature Communications]

Experimental quantification of hydrogen content in the Earth's core

Corresponding Author: Professor Dongyang Huang

Version 0:

Reviewer comments:

Reviewer #1

(Remarks to the Author)

General Comment:

This article presents high quality and innovative research, especially in the methods used, to grant publication in Nature Communications. The results presented here will be valuable for the entire community working on the Earth's core formation.

The experimental strategy and its limitations are well explained, and the conclusion are in line with the results and previous comparable studies.

However, the manuscript could be enhanced, especially the interpretation of the data. In Sections 2.5 and 3, the text is less clear than the rest of the article. A more quantitative approach should be taken: there are a lot of occurrences of 'low concentrations', 'high concentrations' without any reference to an actual value. The calculations that are performed to obtain the range of H concentrations in the core need to be more explicit, at least in an annex, so that other groups who work on the same topic of light elements in the core could do the same calculation and compare their results to this study. In addition, the authors want the reader to believe their method is better than others because they get rid of some assumptions on bulk water content, accretion models and partitioning uncertainties. This is not true, it is a just a way to better hide those assumptions, but they are still there. These parts of the article need to be rewritten somewhat more honestly.

Except for this lack of clarity in the discussion section, most of this article is publishable as it is now.

Moderate comments to be addressed before publication:

First of all, Section 2.3 should be section 3.1: the sections 2.1 and 2.2 are about very experimental and analytical problems, while section 2.3 is mostly about comparing results and proposing interpretations.

On the other hand, Section 3 has no subsection and is significantly shorter than the rest, creating an uneven reading experience. Transforming section 2.3 into section 3.1 and the current section 3 to section 3.2 will greatly improve reader experience (and be more coherent).

L113 to 119: *"because (a) if the diffusion rate" [...] "than one order of magnitude".*

Please give the actual numbers for the diffusion coefficient, and maybe propose a back of the hand calculation to make your point. As it is, your point is not as clear as you think it is. As I understand it, your point is that you are in equilibrium with Si and that H diffusivity being higher than Si, than the H you measure represents the actual concentration. You are concluding that it means that your measurements are valid for Earth like range of P and T. The link between the diffusion coefficient values and the validity of your measurement is tenuous. What kind of value for the diffusivity of H would mean that your results are not applicable to the Earth for instance? Is your point that you do not lose H upon quenching (which you do elsewhere in figure A4 anyway)?

My point is that specialist of metal/silicate equilibrium might be able to reconstruct your logical construction, but other people might not, and the logical links need to be made clearer. Also, make it more quantitative, i.e., give the numerical values you use and express better the calculations you made.

L121-135: *"Hydrogen has long been considered a major light-element candidate" [...] "between studies by one order of magnitude"*

This line of argumentation is confusing. You start by talking about uncertainties in modeling the Earth's H content, then move on to the uncertainties in measuring H in experiments, then move back to model uncertainties while citing experimental studies. It took several readings to understand the point you are making here. I would advise to separate your argument in two:

- one it is difficult to measure H in metal/silicate partitioning, and there are some uncertainties in that, especially by relying on volume change of the iron lattice (but also with Clesi et al. type of measurements);
- two, there is a discussion to have on the initial bulk content during accretion and therefore the availability of water that can go into the core.

Those are two separate questions, which when combined lead to a broader discussion on the H content of the core. Here I am questioning the way you combine those two arguments, which needs to be enhanced.

As it is now, you talk about the limit of volume change determination, then add a "furthermore" and references to DH determination in Li et al., Clesi et al. and Tagawa et al. One would expect a discussion of the limitations on the measures made in those papers, but you actually follow-up with a discussion of the models they proposed. This is really throwing the reader off. As for the previous comment, the logical links here need to be rework to make your point clearer.

L136-138: *"Here, we propose an alternative way, independent of the conventional DH and their associated assumptions/uncertainties, to determine H inventory in Earth's core as follows"*

This sentence is quite problematic, as you are saying you can avoid the assumptions or uncertainties in other studies, but you are not. It is true that the range of Si content in the core is more constrained than the H content, and that the bulk Si content of the Earth is less problematic to estimate (but there are uncertainties there also). However, this range of Si concentrations you rely on to do your calculations are actually relying on the same forward modeling assumptions you disregard (with less uncertainties, but uncertainties nonetheless).

Furthermore, you imply you have a way to estimate the core H content independently of the bulk water content of the Earth and this is simply not true, except for one very specific case: the only way this is an independent method to determine the H content in the core is if, at any point in accretion, the estimated bulk hydrogen content of the planet is enough to reach a H/Si atomic ratio in the core of 1. I am guessing this is the case for the lowest concentrations of Si in the core (~2%wt), but it will become quite harder for scenarios yielding higher concentrations (10 to 12 % wt).

You need to argue for this underlying assumption to your calculation, which is there is enough H to fulfill the requirement to get a value of 1:1 for the H:Si atomic ratio, and this at each step of accretion, by providing the minimal amount of H you need for this to work. Otherwise, if the Earth accreted dry, then your point is mute. If there is not enough H at any point in the core formation process (let's say at the beginning of accretion, when a lot of Si goes into the metallic phase), then your point is mute also. I agree that your method is interesting to estimate the maximum H content of the core, but this is still relying on hidden assumptions about the accretion scenario you chose.

Finally, your own measurements (or rather the one sample you were able to measure, which is quite impressive, but it is still one measurement) also presents uncertainties. I am ready to believe those uncertainties are smaller than other methods, but you still have to take them into account in your calculations, and therefore you are not that much better than other studies (it might be better than classical D_H studies, I will give you that).

These limitations need to be acknowledged properly, or at least these sentences (here or in the abstract) implying you have an "independent" way of estimating the H content in the core, as opposed to the rest of the literature, need to be rewritten. In any case, you cannot escape the discussion about the bulk content of water during accretion, despite removing the water content of the mantle from your equation.

L149-151: *"As such, the Earth's core is estimated to contain 0.07–0.36 wt.% H, corresponding to the H content of 9 to 45 oceans of water."*

We are back on the previous point: you are assuming there is 9 to 45 oceans of water available to go into the core during the entire core formation process, without saying this is an assumption. The Si range varies because of the different assumptions made for accretion process (number of accretion impact, initial composition, oxygen fugacity evolution). Therefore, you are not really escaping the discussion you want to escape, despite finding a method to find the maximum possible H content in the core. My question here is: what is the minimum water content in the bulk Earth and in the silicate Earth to reach those values? You are kind of doing it in the following text, but not very quantitatively or precisely.

Section 3:

This entire section is very much qualitative and is hard to follow. You are mixing two problems (initial water content and early geodynamo). Both problems are complicated enough, and your attempt in linking them is quite confusing from an external point of view. I would advice to be more quantitative and clearer in your logical transitions, because as it is, it is quite complex to follow.

First, you are considering a reasonably moderate amount of water: what is an unreasonably moderate amount? What is reasonably high or low? Then you go on to say that it would make the core the largest reservoir of H: how much is in there (concentrations or bulk mass)? How much is in the mantle and/or the bulk Earth in the same units? Only then we could understand the the core is actually the main reservoir of H.

Then, from L163 to L166, you take the argument backward: you assume there is H in the core and given the unknown concentration you proposed earlier, deduce that the Earth have to be accreted wet. You basically stumbled upon your

hidden assumption from earlier (i.e., there is enough H in the bulk to have H in the core), and what you do here is just circular reasoning. It is because you assumed the Earth accreted wet that there can be H in the core, not the other way around. If this is not your argument, then you definitely need to rewrite and explain more, because this is what I am reading.

Finally, between L167 and 171 you present all the arguments in favor of a wet accretion of the Earth. This is fine, but should be done first, as it is the basis for your H core content determination. You follow-up with a "Furthermore", meaning we expect more arguments to the same point and you completely change the subject. The link between the onset of the geodynamo and the dynamics and isotopic argument for early water accretion is quite mysterious. I think these arguments need to be separated: maybe a new paragraph, and/or get rid of the "Furthermore" which is only misleading here, or at least the implicit link implied by 'furthermore' needs to be made more explicit before.

I also think a bit more details on your point about geodynamo would be we welcomed: by how much (in K) does the presence of hydrogen change the liquidus for instance? Also, you are talking about SiO₂ crystallization and exsolution so I am assuming this is happening at the CMB, but the exsolution and geodynamo start implies more a deep core process. What kind of mechanism and/or model (maybe a qualitative drawing would help here) are you considering to make your point? A little more detail here would go a long way to help the reader understand your point.

Minor comments:

L43-44: "*Large amounts of O*"
What is large? (give a range).

L64: "*To ensure the attainment of equilibrium, the heating durations were between 6 to 10 seconds,*"
Either add the reference you are giving in the methods section here, or send the reader to the methods section, because this is a short time, even for a small sample, to reach equilibrium.

Tables S2 to S5: Please precise which fraction (mass or atomic) you are presenting here.

Reviewer #2

(Remarks to the Author)

This manuscript presents an innovative/state-of-the-art experimental approach to quantifying the hydrogen content of samples in laser-heated diamond anvil cells (LHDACs) combined with atom probe tomography (APT). The direct observation of hydrogen at small Si-O rich nanostructures represents a significant methodological advancement. However, I would like to raise some points for clarification regarding the interpretation of the results and the underlying assumptions of the modeling. I suggest the authors discuss the following limitation.

As a reference for this review, Tagawa et al. (2021) [ref. 11] conducted metal-silicate partitioning experiments for hydrogen under similar P/T conditions, with quantitative Si/H chemical results. I will review this manuscript with reference to that work.

Major Comments:

1. Hydrogen underestimation due to inability to analyze FeH_x

It is well established that Fe reacts with H to form FeH_x at high pressures. However, this FeH_x decomposes when recovered to conditions under 3 GPa at room temperature (this can be confirmed by XRD experiments). Since this study analyzes samples using FIB after quenching to the ambient pressure, any FeH_x present at high pressure-temperature conditions would have completely decomposed by the time of analysis. Therefore, in addition to hydrogen coexisting with Si-O in metal, hydrogen dissolved in the Fe phase likely existed at high pressure-temperature conditions. This critical limitation of the analytical protocol needs to be explicitly stated.

Regarding Line 36, which states that "high-pressure experimental data in measuring H have been suffering from the fact that their H contents were inferred indirectly from the lattice expansion induced by H," I note that there are only three possible methods to measure hydrogen in Fe at high pressure: (1) measurement from lattice volume expansion at high pressure, (2) in situ neutron diffraction at high pressure, or (3) conducting all processes through decompression and APT/SIMS analysis at cryogenic temperatures to prevent H loss.

As a result, if this study can only observe hydrogen in Si-O-rich nanostructures while missing hydrogen in the bulk metal phase, the total hydrogen content in Fe is significantly underestimated. Consequently, the hydrogen content of Earth's core may be underestimated by approximately a factor of two, potentially misleading readers. Therefore, the discussion in Lines 35-38 should be revised to explicitly address FeH_x decomposition during sample preparation, and the conclusions in section 2.3 need to be modified to acknowledge this limitation.

2. Chemical equilibrium between FeH_x and H in nanoscale Si-O-rich quench textures

It is not clarified why the hydrogen content measured in nanoscale Si-O-rich quench textures represents bulk metal hydrogen content at high pressure-temperature conditions. The authors' interpretation relies on three key assumptions:

- 1) H in Si-O-H-rich nanostructures represents all of the hydrogen content on the metal side at high P-T
- 2) Si and H in Si-O-H maintain a 1:1 ratio based on the experimental results in Figure 2b
- 3) The results of 1) and 2) satisfy metal-silicate partitioning equilibrium for both Si and H

Based on this logic, the authors argue that once the Si content in Earth's core is established, the hydrogen content in the core can be determined, yielding estimates of 0.07-0.36 wt% H. It lacks sufficient justification for the following reasons.

Regarding assumption 1): As discussed in major comment 1, hydrogen that would dissolve directly in the Fe lattice (as FeH_x) at high P-T cannot be ignored but is not detected by the present method.

Regarding assumption 2): Chemically rational explanation is needed for why the Si:H ratio should be fixed at 1:1. Without establishing a thermodynamic or structural basis for this ratio, the H content in Si-O-rich phases may vary significantly depending on temperature-pressure conditions. Adding a supplementary figure comparing Si/H ratios to other metal/silicate experiments would also be beneficial. Addressing this point would strengthen the manuscript.

Regarding assumption 3): This study implies that Earth's core hydrogen content can be determined without knowing the metal-silicate partition coefficient for H (D_{H} metal/silicate), which raises several concerns:

- According to this paper's logic, hydrogen follows silicon into the core—but what is the thermodynamic basis for this coupling?

- Are Si and H truly partitioning together, or could they partition independently?

- Doesn't the amount of nanoscale Si-O-rich quench textures vary with experimental temperature, pressure, oxygen fugacity, and cooling rate?

The amount and chemical speciation of Si and O entering Fe varies greatly depending on oxygen fugacity and Si/O partition coefficients. For example, core formation models favor either Si-rich cores (Fischer et al. 2015 GCA) or O-rich cores (Siebert et al. 2013 Science), and Si cannot always exist in the form of SiO₂ in the metal phase.

These unresolved issues regarding the relationship between Si-O-rich textures and bulk hydrogen content weaken the study's quantitative implications about core composition of H. I recommend that the authors make the limitations of their hydrogen estimates clearer and add a more thorough discussion of these uncertainties in section 2.3 and the conclusions.

Minor Comments:

Line 66: Given that metal-silicate partitioning experiments were performed at high temperatures, the samples in the metal phase sometimes exhibited separation. Since APT does not provide information on the large-scale elemental distribution in metal phase, I would appreciate it if the author(s) could present chemical composition mapping data obtained through EDS or other comparable analytical methods.

Fig 1. Caption: I suggest that the discussion regarding D₂ warrants more detailed description in the main text. The spatial proximity of Si and H does not necessarily indicate the formation of SiH ion species. (Considering as high as 4 at% H, cluster ions may have formed.) The experimental results be made more convincing by examining whether the observed ratio is consistent with SiD or with other ion species.

Line 108: The structure of the sample is somewhat unclear for me and would be helpful from a more detailed description. Specifically, are there phases present other than the metallic iron, Si-O-H rich regions within the iron, and the silicate melt?

Line 130: The volume expansion effect induced by Si and O in iron should be discussed quantitatively. Fu et al. (2013) JGR may serve as a valuable reference for this analysis.

Line 167: It would be helpful if the authors could address the possible sources of water to the proto Earth.

Supplementary Table: The SiO₂ content appears high relative to typical MORB glass. Does this affect the experiments?

Despite raising the various points above, I want to emphasize that the ability to analyze the extremely small samples from DAC experiments using FIB-APT is a very significant contribution to geoscience. I express my sincere respect for the authors' efforts in achieving this.

Version 1:

Reviewer comments:

Reviewer #1

(Remarks to the Author)

I have read the changes made by the authors to answer both reviewers comments. As far as I am concerned, the authors have answered all the comments properly and the changes made to the manuscript are sufficient to grant publication. Their response between the two reviewers is quite balanced and I think the paper is much easier and clearer to read than in its previous version. Enough details were added to make it easier for other teams to try and replicate both experimental work and the following calculations.

There may be still some typos (for instance Line 200: example) but, other than that I think no further modifications are necessary.

Reviewer #2

(Remarks to the Author)

General Comments:

This paper presents a direct quantification of H within Si-O-H-rich nanoclusters in quenched iron recovering from core-forming conditions using Atom Probe Tomography (APT). The authors have responded sincerely to all previous comments, and the manuscript has been significantly improved with a much clearer logical structure.

Although metal-silicate partition coefficients of H were not directly calculated from the results, this study provides direct evidence that hydrogen was likely incorporated into the early Earth's core, provided sufficient water was delivered during the early stages of accretion. The addition of Figure 4 effectively clarifies the primary argument, and Figure S1 aids considerably in understanding the experimental settings.

The authors' responses are satisfactory in almost all respects. I believe this work is of high value from both experimental and technical standpoints.

However, to further sharpen the discussion, I recommend addressing the following points, if possible:

1. Distinction between "proto-"core and present core composition

It is necessary to more clearly distinguish between the chemical composition of the initial core and that of the present-day core.

In Section 3.2, the authors discuss the exsolution process that drove the early dynamo over time after core formation. However, since the amounts of Si, O, and H in the core change accordingly, it is currently somewhat unclear whether the text refers to the "present core" or the "past core." So, it might be better to explicitly state in Section 3.1 that these values represent the hydrogen content in the initial core just after formation. Note that such a clarification would not affect the estimate of the total water inventory of the Earth.

Unless the hydrogen content remains largely unchanged following the exsolution of SiO₂, I suggest slightly adjusting the title of Section 3.1 (e.g. "H content in the initial (or proto-) Earth's core.") and the wording in Lines 161–165.

2. Bulk Hydrogen Content in the Metal Phase

The discussion in Lines 203–205 was partially unclear.

As noted in Lines 172–173, assuming that Si-O-H forms at high pressure and the Si-O-H nanostructures subsequently precipitate, there appears to be no clear chemical basis for the Si:H atomic ratio being 1:1 (unity) in molten iron.

As the authors argue in Line 190, this observation may simply reflect a stoichiometric preference where hydrogen is preferentially sequestered into the Si-O-rich nanostructure up to unity H:Si ratio. This implies that the possibility of FeH_x existing separately in the metallic matrix cannot be ruled out, especially if the clusters have already reached their capacity for H at that near-unity ratio.

In that case, the hydrogen content fixed within the Si-O-H structure might actually represent a minimum value rather than the "maximum (upper bound)" claimed in Line 21 of the abstract or Lines 203–205 of the text.

While the Si:H ratio in Figure 4 appears close to 1, the log-scale of the plot might mask variations by a factor that could be significant, (but I think no additional change to Figure 4 is necessary).

As an alternative interpretation consistent with previous studies [e.g., ref 12] where FeH_x, FeOOH_x, and Si were observed with XRD after quenching, the following scenario could be drawn:

- Following quenching, Si-O-H nanostructures precipitate as "blobs" (as shown in Figure S1).
- Since SiO₂ is the core, SiO₂ (stishovite) might be partially visible. (Note that in this experiment, the solid phase precipitating from the glass is Ca-perovskite.)
- Excess O might lead to the formation of FeOOH_x.
- The remaining hydrogen is observed as FeH_x.

Based on these possibilities, I ask the authors to reconfirm whether the descriptions in Line 21 of the abstract and Lines 203–205 of the main text need to be revised. If the authors conclude that no revision is necessary after considering these points, I am willing to respect their judgment.

3. Authors' Responses to Comments from Reviewer #2

Regarding the individual responses to the previous round of reviews, I find them satisfactory and will omit detailed comments, except for the concern raised in Point 2 above. I should mention that the flow of the argument in Lines 215–221—where it is stated that the hydrogen content constrained by this study falls within cosmochemical constraints—felt somewhat difficult to follow. Additionally, for your reference other the hydrogen acquisition model was provided by Olson and Sharp (2018, EPSL).

While I believe the current manuscript is acceptable as it is, I would appreciate it if you could consider points 1–3 above, if possible. I was surprised to learn that it took three years to obtain this data. I express my high respect for the authors' dedication to performing difficult and important study.

Please find below our point-by-point responses. Questions/Comments are in **black**, our reply is in **blue**, and modifications to the manuscript in **red**. All the changes indicated below are incorporated in the revised manuscript.

REVIEWER COMMENTS

Reviewer #1 (Remarks to the Author):

General Comment:

This article presents high quality and innovative research, especially in the methods used, to grant publication in Nature Communications. The results presented here will be valuable for the entire community working on the Earth's core formation.

The experimental strategy and its limitations are well explained, and the conclusion are in line with the results and previous comparable studies.

We wish to thank the reviewer for the kind words, and for recognising our work – thank you very much.

However, the manuscript could be enhanced, especially the interpretation of the data. In Sections 2.5 and 3, the text is less clear than the rest of the article. A more quantitative approach should be taken: there are a lot of occurrences of 'low concentrations', 'high concentrations' without any reference to an actual value.

Thank you for pointing this out. However, we have found only one occurrence of such statement without referring to real values. We have now added the values and the tables to find those values. The relevant sentence in Figure 1 caption now reads:

Apart from Si, O and H, elements that are conventionally lithophile (i.e. Al and Ca) under low to ambient pressure conditions also partition into the metal, albeit at extremely low concentrations (~0.1–0.2 at.%, Tables S2–S3).

The calculations that are performed to obtain the range of H concentrations in the core need to be more explicit, at least in an annex, so that other groups who work on the same topic of light elements in the core could do the same calculation and compare their results to this study.

We have now added Table S7 (copied below), to show how the calculation is done. Thank you for the suggestion.

Terrestrial Reservoir	Surface Ocean	Core	Bulk Silicate Earth (BSE)	Earth
Mass (kg)	1.40E+21	1.91E+24	4.06E+24	5.97E+24
Mass of H ₂ O in BSE				
Reference	H ₂ O concentration in BSE or BE (ppm)	Note	Unit (kg)	Unit (Ocean Mass)
Marty (2012)	1000	H ₂ O in BE (lower bound)	5.97E+21	4
Marty (2012)	3000	H ₂ O in BE (upper bound)	1.79E+22	13
Palme & O'Neil (2014)	1100	H ₂ O in BSE	4.47E+21	3
Hirschmann (2018)	710	H ₂ O in BSE	2.88E+21	2

NB: In the above references, BSE includes hydrospheric inventory.

In addition, the authors want the reader to believe their method is better than others because they get rid of some assumptions on bulk water content, accretion models and partitioning uncertainties. This is not true, it is just a way to better hide those assumptions, but they are still there. These parts of the article need to be rewritten somewhat more honestly.

This summary comment corresponds to the reviewer's below "Moderate comments". We have revised our manuscript accordingly in response to those comments, and we kindly refer the reviewer to our responses in the following texts.

Except for this lack of clarity in the discussion section, most of this article is publishable as it is now.

Thank you very much for your support.

Moderate comments to be addressed before publication:

First of all, Section 2.3 should be section 3.1: the sections 2.1 and 2.2 are about very experimental and analytical problems, while section 2.3 is mostly about comparing results and proposing interpretations.

On the other hand, Section 3 has no subsection and is significantly shorter than the rest, creating an uneven reading experience. Transforming section 2.3 into section 3.1 and the current section 3 to section 3.2 will greatly improve reader experience (and be more coherent).

Thank you for the suggestion. We have now moved Section 2.3 to Section 3.1. It does read better.

L113 to 119: *"because (a) if the diffusion rate" [...] "than one order of magnitude"*.

Please give the actual numbers for the diffusion coefficient, and maybe propose a back of the hand calculation to make your point. As it is, your point is not as clear as you think it is. As I understand it, your point is that you are in equilibrium with Si and that H diffusivity being higher than Si, than the H you measure represents the actual concentration. You are concluding that it means that your measurements are valid for Earth like range of P and T. The link between the diffusion coefficient values and the validity of your measurement is tenuous. What kind of value for the diffusivity of H would mean that your results are not applicable to the Earth for instance? Is your point that you do not lose H upon quenching (which you do elsewhere in figure A4 anyway)?

My point is that specialist of metal/silicate equilibrium might be able to reconstruct your logical construction, but other people might not, and the logical links need to be made clearer. Also, make it more quantitative, i.e., give the numerical values you use and express better the calculations you made.

Thank you for the advice. We have now provided the specific values for the diffusion coefficient, and rephrased this part to better construct our rationale. Lines 113-124 now read:

On the other hand, based on reported diffusivities, one may expect the coupled exsolution of H, O and Si to be valid within the high-pressure and high-temperature domains relevant to Earth-sized core formation. Under probable core-forming conditions (~40–70 GPa, ~3000–4500 K) (Fischer et al., 2015; Huang et al., 2024), the diffusion coefficients in liquid Fe are on the orders of 10^{-8} m²/s for Si and O, and 10^{-7} m²/s for H (Alfe 1999; Huang 2019; Posner 2019). As discussed in Section 1, the Si–O-rich quench texture is an established feature at these P–T conditions in diamond anvil cell experiments; here, what we observed with APT is that H is bound to lock with Si and O. If the diffusion rates of Si and O allow for fast formation of the Si–O-rich nanostructure upon quenching, forming the Si–O–H-rich cluster under hydrous conditions would be inevitable, as the diffusivity of H is higher than those of Si and O by one order of magnitude.

L121-135: *"Hydrogen has long been considered a major light-element candidate" [...] "between studies by one order of magnitude"*

This line of argumentation is confusing. You start by talking about uncertainties in modeling the Earth's H content, then move on to the uncertainties in measuring H in experiments, then move back to model uncertainties while citing experimental studies. It took several readings to understand the point you are making here. I would advise to separate your argument in two:

- one it is difficult to measure H in metal/silicate partitioning, and there are some uncertainties in that, especially by relying on volume change of the iron lattice (but also with Clesi et al. type of measurements);

- two, there is a discussion to have on the initial bulk content during accretion and therefore the availability of water that can go into the core.

Those are two separate questions, which when combined lead to a broader discussion on the H content of the core. Here I am questioning the way you combine those two arguments, which needs to be enhanced.

As it is now, you talk about the limit of volume change determination, then add a "furthermore" and references to DH determination in Li et al., Clesi et al. and Tagawa et al. One would expect a discussion of the limitations on the measures made in those papers, but you actually follow-up with a discussion of the models they proposed. This is really throwing the reader off. As for the previous comment, the logical links here need to be rework to make your point clearer.

We agree with the reviewer that it does take a few careful readings to understand our argument, because, apart from reporting experimental results, some "experimental studies" (e.g. Tagawa et al.) used extensively forward models (e.g. estimated bulk water content) to discuss H content of the core.

To make our argumentation clearer, we take the reviewer's advice and highlight the two sources of uncertainties in inferring H content of Earth's core. Lines 128-144 now read:

For decades, however, our knowledge of the exact content of H in planetary cores is hindered by the inability of unambiguously quantifying H in high pressure samples. There are two main sources of uncertainties in estimating H content in planetary cores.

First, except occasionally H analysis is made available (Clesi et al., 2018; Malavergne et al., 2019), the current practice of estimating H content in the metal phase involves inferring the amount of H alloyed with iron, from the lattice expansion induced by the addition of interstitial H (Iizuka-Oku et al., 2017; Tagawa et al., 2021). This approach, however, relies on accurate determinations of lattice parameters of both iron and iron hydrides, and an inexplicit assumption that simultaneous dissolutions of Si and O in the iron would not induce any lattice expansion of the latter. Although the volume expansion of solid Fe induced by H is found to be approximately eight times greater than that by Si (Fu et al., 2023), the effect of the latter is not negligible. For instance, 9 wt.% Si in hcp Fe at 100 GPa may expand the lattice by about 3% (Tateno et al., 2015). Second, forward models (Clesi et al., 2018; Li et al., 2020; Tagawa et al., 2021) using metal-silicate partition coefficient of H ($D_H \equiv c^{metal}_H / c^{silicate}_H$, where c denotes concentration of H in relevant phase) rely on the estimated water content of the silicate Earth, which further introduces uncertainties as the latter differs between studies by one order of magnitude (Marty et al., 2012; Hirschmann, 2018).

L136-138: *"Here, we propose an alternative way, independent of the conventional DH and their associated assumptions/uncertainties, to determine H inventory in Earth's core as follows"*

This sentence is quite problematic, as you are saying you can avoid the assumptions or uncertainties in other studies, but you are not. It is true that the range of Si content in the core is more constrained than the H content, and that the bulk Si content of the Earth is less problematic to estimate (but there are uncertainties there also). However, this range of Si concentrations you rely on to do your calculations are actually relying on the same forward modeling assumptions you disregard (with less uncertainties, but uncertainties nonetheless).

Furthermore, you imply you have a way to estimate the core H content independently of the bulk water content of the Earth and this is simply not true, except for one very specific case: the only way this is an independent method to determine the H content in the core is if, at any point in

accretion, the estimated bulk hydrogen content of the planet is enough to reach a H/Si atomic ratio in the core of 1. I am guessing this is the case for the lowest concentrations of Si in the core (~2%wt), but it will become quite harder for scenarios yielding higher concentrations (10 to 12 % wt).

You need to argue for this underlying assumption to your calculation, which is there is enough H to fulfill the requirement to get a value of 1:1 for the H:Si atomic ratio, and this at each step of accretion, by providing the minimal amount of H you need for this to work. Otherwise, if the Earth accreted dry, then your point is mute. If there is not enough H at any point in the core formation process (let's say at the beginning of accretion, when a lot of Si goes into the metallic phase), then your point is mute also. I agree that your method is interesting to estimate the maximum H content of the core, but this is still relying on hidden assumptions about the accretion scenario you chose.

Finally, your own measurements (or rather the one sample you were able to measure, which is quite impressive, but it is still one measurement) also presents uncertainties. I am ready to believe those uncertainties are smaller than other methods, but you still have to take them into account in your calculations, and therefore you are not that much better than other studies (it might be better than classical D_H studies, I will give you that).

These limitations need to be acknowledged properly, or at least these sentences (here or in the abstract) implying you have an "independent" way of estimating the H content in the core, as opposed to the rest of the literature, need to be rewritten. In any case, you cannot escape the discussion about the bulk content of water during accretion, despite removing the water content of the mantle from your equation.

Thank you for pointing this out. Indeed, these points should be discussed more thoroughly. We have now expanded our discussions, and elaborated on the point-by-point limitations of our approach. Lines 178-205 now read:

Despite that our approach may avoid certain assumptions/uncertainties in the other studies, it certainly involves uncertainties of its own.

(i) The uncertainties in H quantification using atom probe. As discussed in Section 2.2, the residual H from the chamber may contribute up to 4 at.% of the total measured H. Taken the face value, our measured H content in the Si–O-rich cluster (36 at.%, Table S2) would be overestimated by 11%, which would adjust the H:Si ratio downwardly to 0.97. In Fig. 4, we explore the quantitative effect of varying H:Si ratio on H content of the core.

(ii) The uncertainties in the range of Si content of the core. Although it is generally considered more constrained than the H content, the estimated bulk Si concentration of the core is based on the same forward modelling assumptions. We want to emphasise that the 2–10 wt.% Si is a broad range, and serves only as a reference point.

(iii) The underlying assumption that sufficient amount of H to be present during the course of Earth's accretion. If Earth had accreted relatively 'dry', for example, from mainly enstatite-like material, there would not be enough H available to constantly attain the 1:1 H:Si ratio in the metal core. Hence, considering the above sui generis uncertainties, the estimated amount of 0.07–0.36 wt.% H, or 9–45 oceans of water in Earth's core needs to be treated strictly as an upper bound. As such, the two recent estimates of 0.3–0.6 wt.% (Tagawa et al., 2021) and 0.18–0.49 wt.% (Tsutsumi et al., 2025) likely overestimate the abundance of H in the core.

L149-151: "As such, the Earth's core is estimated to contain 0.07–0.36 wt.% H, corresponding to the H content of 9 to 45 oceans of water."

We are back on the previous point: you are assuming there is 9 to 45 oceans of water available to go into the core during the entire core formation process, without saying this is an assumption. The Si range varies because of the different assumptions made for accretion process (number of accretion impact, initial composition, oxygen fugacity evolution). Therefore, you are not really escaping the discussion you want to escape, despite finding a method to find the maximum possible H content in the core. My question here is: what is the minimum water content in the bulk Earth and in the silicate Earth to reach those values? You are kind of doing it in the following

text, but not very quantitatively or precisely.

Indeed, this is the same point as the previous one. We have now stated the underlying assumption explicitly, and we kindly refer the reviewer to our previous responses.

Regarding the reviewer's question (which is related to one of their "General comments", i.e. the minimum water content required in the bulk Earth), as noticed by the reviewer, we did it in the following text, verbally.

Now, we have provided a new table (Table S7) in the supplementary to show this back of envelope calculation:

Terrestrial Reservoir	Surface Ocean	Core	Bulk Silicate Earth (BSE)	Earth
Mass (kg)	1.40E+21	1.91E+24	4.06E+24	5.97E+24
Mass of H ₂ O in BSE				
Reference	H ₂ O concentration in BSE or BE (ppm)	Note	Unit (kg)	Unit (Ocean Mass)
Marty (2012)	1000	H ₂ O in BE (lower bound)	5.97E+21	4
Marty (2012)	3000	H ₂ O in BE (upper bound)	1.79E+22	13
Palme & O'Neil (2014)	1100	H ₂ O in BSE	4.47E+21	3
Hirschmann (2018)	710	H ₂ O in BSE	2.88E+21	2

NB: In the above references, BSE includes hydrospheric inventory.

Section 3:

This entire section is very much qualitative and is hard to follow. You are mixing two problems (initial water content and early geodynamo). Both problems are complicated enough, and your attempt in linking them is quite confusing from an external point of view. I would advice to be more quantitative and clearer in your logical transitions, because as it is, it is quite complex to follow.

First, you are considering a reasonably moderate amount of water: what is an unreasonably moderate amount? What is reasonably high or low? Then you go on to say that it would make the core the largest reservoir of H: how much is in there (concentrations or bulk mass)? How much is in the mantle and/or the bulk Earth in the same units? Only then we could understand the the core is actually the main reservoir of H.

These are essentially the same points as raised in the last comment. We kindly refer the reviewer to the new Table S7, which has been copied in the above lines.

Then, from L163 to L166, you take the argument backward: you assume there is H in the core and given the unknown concentration you proposed earlier, deduce that the Earth have to be accreted wet. You basically stumbled upon your hidden assumption from earlier (i.e., there is enough H in the bulk to have H in the core), and what you do here is just circular reasoning. It is because you assumed the Earth accreted wet that there can be H in the core, not the other way around. If this is not your argument, then you definitely need to rewrite and explain more, because this is what I am reading.

Indeed. Thanks for pointing this out. This part has been rephrased in the revised manuscript, lines 211-215 now read:

As discussed earlier, such amount of water relies on the assumption that Earth mostly accreted water during main stages of its accretion (Halliday & Canup, 2022), similar to other major volatiles such as C (Fischer et al., 2020) and N (Huang et al., 2024), instead of late delivery through hydrated chondritic materials. This is in line with the non-chondritic (cf. CI chondrite) H, C and N ratios observed in the bulk silicate Earth (Marty, 2012).

Finally, between L167 and 171 you present all the arguments in favor of a wet accretion of the Earth. This is fine, but should be done first, as it is the basis for your H core content determination. You follow-up with a "Furthermore", meaning we expect more arguments to the same point and you completely change the subject. The link between the onset of the

geodynamo and the dynamics and isotopic argument for early water accretion is quite mysterious. I think these arguments need to be separated: maybe a new paragraph, and/or get rid of the “Furthermore” which is only misleading here, or at least the implicit link implied by ‘furthermore’ needs to be made more explicit before.

I also think a bit more details on your point about geodynamo would be we welcomed: by how much (in K) does the presence of hydrogen change the liquidus for instance? Also, you are talking about SiO₂ crystallization and exsolution so I am assuming this is happening at the CMB, but the exsolution and geodynamo start implies more a deep core process. What kind of mechanism and/or model (maybe a qualitative drawing would help here) are you considering to make your point? A little more detail here would go a long way to help the reader understand your point.

Thank you for the suggestion. We have now replaced the “Furthermore” with a new paragraph, to discuss in more detail the implications for the geodynamo. However, at this stage, we restrict our discussion to qualitative aspects, because, importantly, quantitative models require substantive new data (e.g. detailed phase diagram of the Fe-Si-O-H quaternary at CMB pressures), which necessitates much more experimental/theoretical work in the future and is therefore beyond the scope of this manuscript. The new paragraph in lines 222-240 reads:

The coupled sequestration of Si, O and H in core-forming metals at high pressures have important implications for geodynamics, geochemistry and water cycle in deep Earth. While the details of deep-Earth water cycle could be refined with an accurate deep-time geotherm and Fe–Si–O–H phase diagram, the overarching mechanism of core–mantle water transport, driven by coupled dissolution and exsolution of Si, O and H in liquid Fe-rich alloys, remains robust. First, the strong affinity of H with the Si–O-rich nanostructure suggests that similar crystalline H–Si–O solids would form during cooling in the Fe–Si–O–H quaternary. Second, the addition of H would presumably delay the crystallisation of SiO₂ [32] during secular cooling of Earth’s core, via lowering its melting temperature. Early dynamo, if partly driven by core exsolutions, might then require an alternative power source [18, 56], and may have to take account of the effect of H. Subsequently, if, depending on the cooling rate of the liquid core, H–Si–O phases crystallise and ascend to the base of the mantle, the buoyancy and latent heat produced by this crystallisation would promote convection in both the core and overlying mantle. Should it interact with deep-rooted mantle plumes [57], it may imprint the latter with primordial isotopic signatures preserved in the core since its formation [58–60]. Finally, the exsolution of H–Si–O solids would release the early core-sequestered water into the mantle, thereby reshaping mantle rheology, melting behaviour, and our understanding of Earth’s deep water cycle.

Minor comments:

L43-44: “Large amounts of O”
What is large? (give a range).

1–17 wt.%, which has been added in line 44.

L64: “To ensure the attainment of equilibrium, the heating durations were between 6 to 10 seconds,”

Either add the reference you are giving in the methods section here, or send the reader to the methods section, because this is a short time, even for a small sample, to reach equilibrium.

We have now referred the reader to both the references and the Method section:

(Huang et al., 2021; see Methods)

Tables S2 to S5: Please precise which fraction (mass or atomic) you are presenting here.

It is atomic, which is now expressed explicitly in all the tables. Thank you for the advice.

Reviewer #2 (Remarks to the Author):

This manuscript presents an innovative/state-of-the-art experimental approach to quantifying the hydrogen content of samples in laser-heated diamond anvil cells (LHDACs) combined with atom probe tomography (APT). The direct observation of hydrogen at small Si-O rich nanostructures represents a significant methodological advancement. However, I would like to raise some points for clarification regarding the interpretation of the results and the underlying assumptions of the modeling. I suggest the authors discuss the following limitation.

As a reference for this review, Tagawa et al. (2021) [ref. 11] conducted metal-silicate partitioning experiments for hydrogen under similar P/T conditions, with quantitative Si/H chemical results. I will review this manuscript with reference to that work.

Thank you very much for recognising our endeavour. We try our best to address the comments as follows.

Major Comments:

1. Hydrogen underestimation due to inability to analyze FeH_x

It is well established that Fe reacts with H to form FeH_x at high pressures. However, this FeH_x decomposes when recovered to conditions under 3 GPa at room temperature (this can be confirmed by XRD experiments). Since this study analyzes samples using FIB after quenching to the ambient pressure, any FeH_x present at high pressure-temperature conditions would have completely decomposed by the time of analysis. Therefore, in addition to hydrogen coexisting with Si-O in metal, hydrogen dissolved in the Fe phase likely existed at high pressure-temperature conditions. This critical limitation of the analytical protocol needs to be explicitly stated.

Regarding Line 36, which states that "high-pressure experimental data in measuring H have been suffering from the fact that their H contents were inferred indirectly from the lattice expansion induced by H," I note that there are only three possible methods to measure hydrogen in Fe at high pressure: (1) measurement from lattice volume expansion at high pressure, (2) in situ neutron diffraction at high pressure, or (3) conducting all processes through decompression and APT/SIMS analysis at cryogenic temperatures to prevent H loss.

As a result, if this study can only observe hydrogen in Si-O-rich nanostructures while missing hydrogen in the bulk metal phase, the total hydrogen content in Fe is significantly underestimated. Consequently, the hydrogen content of Earth's core may be underestimated by approximately a factor of two, potentially misleading readers. Therefore, the discussion in Lines 35-38 should be revised to explicitly address FeH_x decomposition during sample preparation, and the conclusions in section 2.3 need to be modified to acknowledge this limitation.

We fully agree with the reviewer in that, in the Fe-H binary system, FeH_x forms upon quenching, and decomposes at ambient pressure. However, the picture in the Fe-Si-O-H quaternary system is much less clear than it seemed, because no phase diagram for this quaternary is available under core-forming pressures at the moment.

Nevertheless, we take the reviewer's comment seriously, and discuss this important point in lines 185-194:

Another uncertainty comes from potential underestimation of H content in the Fe matrix. Although H is found to have strong affinity with the Si-O-rich nanocluster, the exact partition coefficient of H between the Si-O-rich nanocluster and Fe matrix is unknown under these pressures, which

deserves serious investigation in future studies. As such, residual H in the matrix may form iron hydride (FeH_x) with Fe upon quenching, in which H is capable of escaping from solid iron during decompression. A recent study estimated H content of Earth's core, based on lattice expansion of Fe introduced by the addition of H (Tagawa et al., 2021). If one takes their surface values, i.e. 0.18–0.49 wt.% H in the core, our values of 0.07–0.36 wt.% H would be underestimated by a factor of ~ 2 at most, provided most of the H does not enter the Si-O-rich nanocluster.

2. Chemical equilibrium between FeH_x and H in nanoscale Si-O-rich quench textures

It is not clarified why the hydrogen content measured in nanoscale Si-O-rich quench textures represents bulk metal hydrogen content at high pressure-temperature conditions. The authors' interpretation relies on three key assumptions:

- 1) H in Si-O-H-rich nanostructures represents all of the hydrogen content on the metal side at high P-T
- 2) Si and H in Si-O-H maintain a 1:1 ratio based on the experimental results in Figure 2b
- 3) The results of 1) and 2) satisfy metal-silicate partitioning equilibrium for both Si and H

Based on this logic, the authors argue that once the Si content in Earth's core is established, the hydrogen content in the core can be determined, yielding estimates of 0.07-0.36 wt% H. It lacks sufficient justification for the following reasons.

Thank you for the comments. We will address these 3 points one by one as follows.

Regarding assumption 1): As discussed in major comment 1, hydrogen that would dissolve directly in the Fe lattice (as FeH_x) at high P-T cannot be ignored but is not detected by the present method.

As noted by the reviewer, this comment “regarding assumption 1)” is essentially the same as “major comment 1”. We kindly refer the reviewer to our responses in previous lines.

Regarding assumption 2): Chemically rational explanation is needed for why the Si:H ratio should be fixed at 1:1. Without establishing a thermodynamic or structural basis for this ratio, the H content in Si-O-rich phases may vary significantly depending on temperature-pressure conditions. Adding a supplementary figure comparing Si/H ratios to other metal/silicate experiments would also be beneficial. Addressing this point would strengthen the manuscript.

The reviewer is right in questioning the possible change of the H/Si ratio in response to P-T conditions. We tried very hard to quantify this at varying P-T conditions, but succeeded with only one sample, due to the extremely high failure rate of APT preparation, particularly with this type of sample (Table S1).

However, we did take seriously the reviewer's comment, and relaxed the molar H/Si ratio, from 1 to a range of 0.1 to 10. Such a range should suffice to show the effect of varying H:Si that may occur at different P-T conditions. To illustrate this, we have added Figure 4 in the revised manuscript as follows.

Fig. 4 Metal–silicate partition coefficient of hydrogen inferred from varying H/Si ratios in the metal. The APT-observed H/Si ratio is close to unity (see the main text), which is allowed in here to vary from 0.1 to 10, in order to quantify its effect on D_H . D_H (dashed lines) is calculated using (i) an Si content of the core at given pressures as determined by Si partitioning model in Siebert et al., (2013), and (ii) a moderate H content (120 ppm) in the primitive mantle (Palme & O’Neil, 2014).

Regarding assumption 3): This study implies that Earth's core hydrogen content can be determined without knowing the metal-silicate partition coefficient for H (D_H metal/silicate), which raises several concerns:

- According to this paper's logic, hydrogen follows silicon into the core—but what is the thermodynamic basis for this coupling?
- Are Si and H truly partitioning together, or could they partition independently?

It is clear that we didn't express ourselves well in the manuscript. We apologise for this, and would like to take the opportunity to improve on it. Lines 169-174 now read:

We would also like to stress that the use of the H/Si ratio to infer H content does not necessarily imply that H follows Si into the metal during metal–silicate partitioning. There are two processes involved in here: metal–silicate equilibration at high temperatures, and exsolution of the Si–O–H-rich nanostructure in liquid Fe upon quenching. While the APT-observed results indicate the latter, it does not discern specific mechanisms occurred in the former.

- Doesn't the amount of nanoscale Si-O-rich quench textures vary with experimental temperature, pressure, oxygen fugacity, and cooling rate?

To be honest, we do not have the answer to this question yet. The current data collected at one P-T condition took us nearly three years. What we can say, however, based on all the published experimental work on partitioning, is that the formation of the Si-O-rich quench texture is unavoidable under core-forming P-T conditions. Such examples can be easily found in the body of LHDAC work during the past decade or so.

We hope that, although the experimental success rate is not high, our work would entail, instead of discourage, more effort from the community to tackle this challenging subject in the future.

The amount and chemical speciation of Si and O entering Fe varies greatly depending on oxygen fugacity and Si/O partition coefficients. For example, core formation models favor either Si-rich cores (Fischer et al. 2015 GCA) or O-rich cores (Siebert et al. 2013 Science), and Si cannot always exist in the form of SiO_2 in the metal phase.

The reviewer is correct in that the partitioning of Si and O varies a lot with oxygen fugacity. In fact, it is precisely the different fO_2 models chosen in Fischer's and Siebert's papers that make the difference.

Fischer's Si-rich core is based on a single-stage core formation model anchored at 54 GPa, and becomes widely cited as it appears in their abstract. However, if one reads the paper carefully, in their Fig. 8A, using the same Si partitioning model, Fischer predicts an O-rich core, when a slightly more oxidised fO_2 path is chosen.

However, we agree that this aspect should be discussed thoroughly, so we now add it into lines 147-152:

It has been established that O partitioning into the metal is greatly enhanced at increasing pressures and temperatures (see Section 1). An O-rich core would readily satisfy the exsolution of the Si-O-rich nanostructure, and is therefore more relevant to Earth-sized planets (Huang & Dorn, 2025). While the exact core composition remains highly debated, a recent review, making use of mineral physics and cosmochemical constraints, favours a slightly more O-rich core (Hirose et al., 2021).

These unresolved issues regarding the relationship between Si-O-rich textures and bulk hydrogen content weaken the study's quantitative implications about core composition of H. I recommend that the authors make the limitations of their hydrogen estimates clearer and add a more thorough discussion of these uncertainties in section 2.3 and the conclusions.

Indeed – we hope in the above responses, we have addressed various points raised, and have made the limitations of our approach clearer. Thank you again for these comments.

Minor Comments:

Line 66: Given that metal-silicate partitioning experiments were performed at high temperatures, the samples in the metal phase sometimes exhibited separation. Since APT does not provide information on the large-scale elemental distribution in metal phase, I would appreciate it if the author(s) could present chemical composition mapping data obtained through EDS or other comparable analytical methods.

The reviewer is correct in mentioning the large-scale analysis in the metal phase. Ideally, one should be able to provide compositional information on both nanoscale and micron scale. However, our own experience is that, one simply cannot have both, particularly in the case of H quantification with APT. Because both EDS and EPMA requires the FIB-recovered sample to be metalised by C (or Pt, Au, etc.), in order to be analysed. This introduces severe sample contamination that would largely compromise the APT results. Especially the EPMA analysis, which is a separate machine and therefore requires the sample to be taken out of the FIB vacuum chamber before the analysis. When exposed to the air (however short the duration), such a small apex (~20 nm in diameter) will absorb water, which again would compromise the APT data. This is why we kept the apex in vacuum after FIB preparation, and enabled cryo-transfer to minimise such contaminations whenever possible (see Methods, Section 4.3).

On the other hand, we share the same concern with the reviewer on this. To convince the reviewer, as much as to convince ourselves, we went back to the literature to check if there were any other phase separation in the metal phase, apart from the Fe matrix and the Si-O-rich nanostructures, and we found nothing but the same feature. We have made this new supplementary figure to show this repeatedly observed morphology produced among various labs:

Fig. S1. Si–O-rich nanostructure formed within liquid iron-rich alloys during quenching in laser-heated diamond anvil cell experiments. (a–d) Backscattered electron images from the literature [15, 20–27], showing the Si–O-rich quench texture (darker ‘bubbles’, typically < 200 nm) embedded within liquid iron-rich alloys (brighter matrix). These alloys previously equilibrated with molten silicates, following the partitioning of Si and O from silicate melts (see the main text). This repeatedly observed nanostructure has now been sampled and analysed, for the presence of H, using atom probe tomography, as reported in Figs. 1–3.

Fig 1. Caption: I suggest that the discussion regarding D2 warrants more detailed description in the main text. The spatial proximity of Si and H does not necessarily indicate the formation of SiH ion species. (Considering as high as 4 at% H, cluster ions may have formed.) The experimental results be made more convincing by examining whether the observed ratio is consistent with SiD or with other ion species.

First, there are no D2 peaks being observed in Fig. 1, only H⁺ and D⁺ (or equivalently H₂⁺, which APT is unable to discern). It is true that the spatial proximity of Si and H does not guarantee the field evaporation of SiH⁺ ions, 4 at% residual H would also do. However, it would be highly unlikely that the residual H (4 at% at most) would field-evaporated into SiH⁺ ions that comprise > 30 at% of the total ions.

If we understand correctly, the reviewer wanted us to strengthen our argument by comparing with SiD or other ion species. While we did not observe any SiD peaks, there are other peaks that could be used to substantiate the H quantification. Lines 96-99 now read:

On the other hand, the observation of endogenous H is supported by the relatively high abundances of protonated major rock-forming elements, e.g. CaH⁺ and SiOH⁺ at Da = 41 and 45, respectively (Fig. 1a).

Line 108: The structure of the sample is somewhat unclear for me and would be helpful from a more detailed description. Specifically, are there phases present other than the metallic iron, Si-O-H rich regions within the iron, and the silicate melt?

This comment is in essence the same as the reviewer’s first “minor comment”. We kindly refer the reviewer to our earlier response to this.

Line 130: The volume expansion effect induced by Si and O in iron should be discussed quantitatively. Fu et al. (2013) JGR may serve as a valuable reference for this analysis.

Thank you for the suggestion and the reference. We have read carefully this paper, and added more thoughts on this in lines 137-140:

Although the volume expansion of solid Fe induced by H is found to be approximately eight times greater than that by Si (Fu et al., 2023), the effect of the latter is not negligible. For instance, 9 wt.% Si in hcp Fe at 100 GPa may expand the lattice by about 3% (Tateno et al., 2015).

Line 167: It would be helpful if the authors could address the possible sources of water to the proto Earth.

Thank you for the comment. We now rephrase this part to better address possible sources of water in Earth. Lines 215-221 now read:

That the Earth accreted most of its water in situ along its formation, is plausible from the dynamical point of view regarding source regions of terrestrial water (Morbidelli et al., 2000). It is also consistent with that the Earth may have been built mainly from enstatite chondrite-like materials, which, apart from their isotopic similarities to the Earth (Javoy et al., 2010; Dauphas, 2017)), contain sufficient amounts of hydrogen to deliver more than three oceans of water and Earth-like H isotopic signatures (Piani et al., 2020).

Supplementary Table: The SiO₂ content appears high relative to typical MORB glass. Does this affect the experiments?

The SiO₂ content seems high, because our MORB glass were synthesised on an FeO-free and minor element-free basis, while fixing the major oxides to their MORB ratios. This should not affect the experiments, because silicate melt composition controls partitioning behaviour through the degree of polymerisation in the melt (cf. Huang et al., GCA, 2021), which is determined by the ratios of major oxides. As long as the major oxide ratios are kept MORB like, it should not affect the experiments.

Despite raising the various points above, I want to emphasize that the ability to analyze the extremely small samples from DAC experiments using FIB-APT is a very significant contribution to geoscience. I express my sincere respect for the authors' efforts in achieving this.

Thank you again for the kind support.

Sincerely,
Dongyang Huang

Please find below our point-by-point responses. Questions/Comments are in **black**, our reply is in **blue**, and modifications to the manuscript in **red**. All the changes indicated below are incorporated in the revised manuscript.

REVIEWERS' COMMENTS

Reviewer #1 (Remarks to the Author):

I have read the changes made by the authors to answer both reviewers comments. As far as I am concerned, the authors have answer all the comments properly and the changes made to the manuscript are sufficient to grant publication. Their response between the two reviewers is quite balanced and I think the paper is much easier and clearer to read than in its previous version. Enough details were added to make it easier for other teams to try and replicate both experimental work and the following calculations.

There may be still some typos (for instance Line 200: exmaple) but, other than that I think no further modifications are necessary.

Thank you very much for the kind support. We have carefully reviewed the manuscript and corrected the remaining typos.

Reviewer #2 (Remarks to the Author):

General Comments:

This paper presents a direct quantification of H within Si-O-H-rich nanoclusters in quenched iron recovering from core-forming conditions using Atom Probe Tomography (APT). The authors have responded sincerely to all previous comments, and the manuscript has been significantly improved with a much clearer logical structure.

Although metal-silicate partition coefficients of H were not directly calculated from the results, this study provides direct evidence that hydrogen was likely incorporated into the early Earth's core, provided sufficient water was delivered during the early stages of accretion. The addition of Figure 4 effectively clarifies the primary argument, and Figure S1 aids considerably in understanding the experimental settings.

The authors' responses are satisfactory in almost all respects. I believe this work is of high value from both experimental and technical standpoints.

We are grateful for the reviewer's recognition of our work – thank you very much.

However, to further sharpen the discussion, I recommend addressing the following points, if possible:

1. Distinction between "proto-"core and present core composition

It is necessary to more clearly distinguish between the chemical composition of the initial core and that of the present-day core.

In Section 3.2, the authors discuss the exsolution process that drove the early dynamo over time after core formation. However, since the amounts of Si, O, and H in the core change accordingly, it is currently somewhat unclear whether the text refers to the "present core" or the "past core." So, it might be better to explicitly state in Section 3.1 that these values represent the hydrogen content in the initial core just after formation. Note that such a clarification would not affect the estimate of the total water inventory of the Earth.

Unless the hydrogen content remains largely unchanged following the exsolution of SiO_2 , I suggest slightly adjusting the title of Section 3.1 (e.g. "H content in the initial (or proto-) Earth's core.") and the wording in Lines 161–165.

Excellent point. The title of Section 2.3 (previously Sec. 3.1, because the journal does not allow for subheadings in the Discussion section) now reads:

H content in Earth's proto-core

And lines 163–165 reads:

As such, Earth's proto-core, i.e. the core after its formation but prior to potential mass loss to the mantle (cf. Section 3.2), is estimated to contain 0.07–0.36 wt.% H, corresponding to the H content of 9 to 45 oceans of water.

2. Bulk Hydrogen Content in the Metal Phase

The discussion in Lines 203–205 was partially unclear.

As noted in Lines 172–173, assuming that Si-O-H forms at high pressure and the Si-O-H nanostructures subsequently precipitate, there appears to be no clear chemical basis for the Si:H atomic ratio being 1:1 (unity) in molten iron.

As the authors argue in Line 190, this observation may simply reflect a stoichiometric preference where hydrogen is preferentially sequestered into the Si-O-rich nanostructure up to unity H:Si ratio. This implies that the possibility of FeH_x existing separately in the metallic matrix cannot be ruled out, especially if the clusters have already reached their capacity for H at that near-unity ratio.

In that case, the hydrogen content fixed within the Si-O-H structure might actually represent a minimum value rather than the "maximum (upper bound)" claimed in Line 21 of the abstract or Lines 203–205 of the text.

While the Si:H ratio in Figure 4 appears close to 1, the log-scale of the plot might mask variations by a factor that could be significant, (but I think no additional change to Figure 4 is necessary).

As an alternative interpretation consistent with previous studies [e.g., ref 12] where FeH_x, FeOOH_x, and Si were observed with XRD after quenching, the following scenario could be drawn:

- Following quenching, Si-O-H nanostructures precipitate as "blobs" (as shown in Figure S1).
- Since SiO₂ is the core, SiO₂ (stishovite) might be partially visible. (Note that in this experiment, the solid phase precipitating from the glass is Ca-perovskite.)
- Excess O might lead to the formation of FeOOH_x.
- The remaining hydrogen is observed as FeH_x.

Based on these possibilities, I ask the authors to reconfirm whether the descriptions in Line 21 of the abstract and Lines 203–205 of the main text need to be revised. If the authors conclude that no revision is necessary after considering these points, I am willing to respect their judgment.

Thank you for pointing this out. Indeed, our results cannot rule out the possibility of FeH_x existing separately in the metallic matrix. We have rewritten lines 202–208 to explicitly acknowledge this point:

Hence, the estimated amount of 0.07–0.36 wt.% H, or 9–45 oceans of water in Earth's core should be interpreted cautiously in light of the above sui generis uncertainties. On the other hand, had Earth accreted relatively 'wet', sufficiently so not only to attain the 1:1 H:Si ratio in the proto-core, but also to allow for the formation of FeH_x following the exsolution of the Si-O-H-rich phase during cooling, H content of the proto-core may be adjusted upward to potentially overlap with the two recent estimates of 0.3–0.6 wt.% (Tagawa et al., 2021) and 0.18–0.49 wt.% (Tsutsumi et al., 2025).

Accordingly, we have removed 'a maximum of' in line 21 in the Abstract, which now reads:

Earth's core is estimated to contain 0.07–0.36 wt.% H, equivalent to 9–45 oceans of water.

3. Authors' Responses to Comments from Reviewer #2

Regarding the individual responses to the previous round of reviews, I find them satisfactory and will omit detailed comments, except for the concern raised in Point 2 above. I should mention that the flow of the argument in Lines 215–221—where it is stated that the hydrogen content constrained by this study falls within cosmochemical constraints—felt somewhat difficult to follow. Additionally, for your reference other the hydrogen acquisition model was provided by Olson and Sharp (2018, EPSL).

Indeed, it is not super clear what we tried to say. We have rephrased it, added the reference suggested by the reviewer – lines 218–222 now read:

That the Earth accreted most of its water in situ along its formation is plausible from a dynamical point of view, with water delivered by planetesimals and planetary embryos (Morbidelli et al., 2000), and is consistent with a hydrogen ingassing model involving interaction between a primordial atmosphere and a magma ocean (Olson and Sharp, 2018).

While I believe the current manuscript is acceptable as it is, I would appreciate it if you could consider points 1–3 above, if possible. I was surprised to learn that it took three years to obtain this data. I express my high respect for the authors' dedication to performing difficult and important study.

Thank you again for the kind words.

Sincerely,
Dongyang Huang